# Orientation Growth of N-Doped and Iron-Based Metal–Organic Framework and Its Application for Removal of Cr(VI) in Wastewater

**DOI:** 10.3390/molecules29051007

**Published:** 2024-02-26

**Authors:** Yan Chen, Chao Lei, Yong-Gang Zhao, Ming-Li Ye, Kun Yang

**Affiliations:** 1Department of Environmental Science, Zhejiang University, Hangzhou 310058, China; cyrosemary@hotmail.com (Y.C.); kyang@zju.edu.cn (K.Y.); 2Polytechnic Institute, Zhejiang University, Hangzhou 310027, China; 3College of Biological and Environmental Engineering, Zhejiang Shuren University, Hangzhou 310015, China; chaolei_212@163.com

**Keywords:** magnetic metal–organic framework (magnetic MOFs), adsorption, Cr(VI), ammonium hydroxide, wastewater

## Abstract

A series of NH_2_-functionalized nano-sized magnetic metal–organic frameworks (MOFs) were prepared in this study for Cr(VI) removal from wastewater. It was observed that not only the morphological, i.e., orientation growth of N-doped and iron-based metal–organic frameworks, but also the adsorption of magnetic MOFs is largely related to the used amount of ammonium hydroxide in preparation. For example, with increasing amounts of ammonium hydroxide used in preparation, the morphology of magnetic MOFs changed from spherical to cube and triangular cone. Moreover, the maximum adsorption capacity of spherical-magnetic MOFs, cubic-magnetic MOFs and triangular cone-magnetic MOFs could be up to 204.08 mg/g, 232.56 mg/g and 270.27 mg/g, respectively. Under optimal conditions, the adsorption process of magnetic MOFs for Cr(VI) was consistent with the pseudo-second-order rate equation (R^2^ = 1) and Langmuir isotherm model (R^2^ > 0.99). Therefore, magnetic MOFs developed in this work offered a viable option for the removal of Cr(VI) from wastewater.

## 1. Introduction

With the rapid development of global industrialization (electroplating processes, pigment industry, tanning of leather), illicit discharge of wastewater containing Cr(VI) ions has posed an increasing danger to people’s health [1,2,3,4]. Cr(VI) ions are liable to be absorbed through skin, digestive and mucous membranes, causing vomiting, dermatitis, eczema and abdominal pain [5,6,7,8]. The main risk is the possibility of cancer from inhalation or long-term exposure to Cr(VI) [9,10], and so Cr(VI) ions have been identified as a priority toxic pollutant by the U.S. EPA [11]. It is reported that Cr(VI) is one of the most frequently detected heavy metals in wastewater, with common concentrations of 50~100 mg/L, which is much higher than the maximum allowed concentration in drinking water (0.05 mg/L) [7]. Therefore, it is essential to study Cr(VI) removal from water to solve heavy metal pollution.

For the precaution and removal of Cr(VI), many treatments have been investigated and reported, including adsorption, oxidation–reduction, ion exchange, deposition and photocatalysis [12,13,14,15,16,17]. Among these methods, adsorption effectively concentrates Cr(VI) by two main forces, i.e., electrostatic interactions and coordination interactions, which are recognized as easy, fast, and highly efficient environmental protection techniques without secondary waste [18,19,20,21,22,23]. A wide variety of adsorption materials for the removal of Cr(VI) in wastewater have been reported, such as metal hydroxides [24], silica-based materials [25,26,27], biochar [28], activated carbon [29,30,31], various types of clay minerals [32,33], chitin and chitosan [34,35,36], zeolites [37], graphene oxide [38,39,40], amino-functionalized silica [41], amino-functionalized magnetic polymers [42] and metal–organic frameworks (MOFs) [43,44,45,46]. Among these adsorption materials, MOFs are a novel class of crystalline porous material with periodic network structure formed by the interconnection of inorganic metal centers (metal ions or metal clusters) and bridging organic ligands through self-assembly [43,44,45]. As novel organic–inorganic hybrid materials, MOFs are different from inorganic porous materials and general organic complexes. They combine the rigidity of inorganic materials with the flexibility of organic materials. MOFs with typical architectures and interesting topologies have been designed and used as novel adsorbents for Cr(VI) removal in wastewater. Compared with other traditional adsorption materials, MOFs show much higher thermal stability, sensitivity and selectivity [47,48,49,50,51]. Specifically, the adsorption process is carried out in the crystalline state of MOFs, reducing the loss of adsorbents and recycling of the materials. For example, Fei et al. reported a cationic metal–organic material, which was used for Cr(VI) removal, and the adsorption capacity was 68.5 mg/g [52]. In order to improve the adsorption capacity of MOFs to Cr(VI), Shao et al. developed a novel water-stable Cu(II)-MOF material, which was successfully used for Cr(VI) removal from water [53]. The results show that Cu(II)-MOF material efficiently adsorbs Cr(VI) ions via a single-crystal to single-crystal coordination substitution process, and the adsorption capacity of Cu(II)-MOF to Cr(VI) was 190 mg/g. Compared with cationic-MOF, although the adsorption capacity of Cu(II)-MOF material has been greatly improved, the difficulty in the rapid separation of MOFs from water samples has limited further application. Thus, developing more suitable MOFs has become a new research hotspot.

For magnetic materials, Hu et al. developed γ-Fe_2_O_3_ used for Cr(VI) removal in wastewater. γ-Fe_2_O_3_ has superparamagnetic features, which can be separated quickly from water samples under a magnetic field, while its adsorption capacity for Cr(VI) was just 17.0 mg/g [54]. In our previous work, ethylenediamine-functionalized magnetic polymers (EDA-MPs) have been reported to remove Cr(VI) from wastewater. However, the maximum adsorption capacities of EDA-MPs were 61.35 mg/g, which is much lower than the MOFs [55]. Therefore, combining the best of both magnetic materials and MOFs inspired a novel magnetic MOF with EDA as the organic ligand, which would likely become a powerful adsorbent for Cr(VI) removal in wastewater.

In the present study, a kind of NH_2_-functionalized nano-sized magnetic MOF with different morphology, named spherical-magnetic MOFs, cubic-magnetic MOFs and triangular cone-magnetic MOFs, were prepared. They were characterized by scanning electron microscopy (SEM), X-ray diffractometer (XRD), vibrating sample magnetometer (VSM) and X-ray photoelectron spectroscopy (XPS). The adsorption performances of the magnetic MOFs for the removal of Cr(VI) was studied based on batch tests.

## 2. Results and Discussion

### 2.1. Characterization of NH_2_-Functionalized Nano-Sized Magnetic MOFs

#### 2.1.1. SEM and XRD Analyses of NH_2_-Functionalized Nano-Sized Magnetic MOFs

The SEM images of the NH_2_-functionalized nano-sized magnetic MOFs, i.e., spherical-magnetic MOFs, cubic-magnetic MOFs and triangular cone-magnetic MOFs, are shown in Figure 1. It is revealed that the morphologies of spherical-magnetic MOFs without using ammonium hydroxide in the preparation are spherical with a diameter of 50 nm. Meanwhile, with an increased amount of ammonium hydroxide used in the preparation procedure, the morphology of magnetic MOFs is changed to cubic and triangular cone, respectively. The XRDs of magnetic MOFs are shown in Figure 2, which was considered the representative for the discussion. According to Figure 2, six characteristic peaks of Fe_3_O_4_ at 2*θ* of 30.1°, 35.5°, 43.1°, 53.4°, 57.0° and 62.6° were found in the XRD patterns of three kinds of magnetic MOFs, i.e., spherical-magnetic MOFs, cubic-magnetic MOFs and triangular cone-magnetic MOFs, which corresponded to the indices (220), (311), (400), (422), (511) and (400). Furthermore, peaks of Fe_4_N at 2*θ* of 7.3°, 8.2°, 43.3°, 50.5° and 74.6° were found in the XRD patterns of three kinds of magnetic MOFs, which corresponded to indices (111), (002), (110), (200) and (220), and this result was consistent with the literature [56]. It can be concluded that the modification of Fe_3_O_4_ nanoparticles did not make measurable changes in the phase property of magnetic cores, and the Fe-N of magnetic MOFs was formed.

#### 2.1.2. VSM Analysis of NH_2_-Functionalized Nano-Sized Magnetic MOFs

The paramagnetic performances of the NH_2_-functionalized nano-sized magnetic MOFs were analyzed by VSM, as shown in Figure 3. It can be seen that the saturation moments were found to be 48.54, 52.46 and 54.48 emu/g for spherical-magnetic MOFs, cubic-magnetic MOFs and triangular cone-magnetic MOFs, respectively. It is revealed that saturation moments of spherical-magnetic MOFs without using ammonium hydroxide in the preparation are low, while with increasing the amount of ammonium hydroxide used in the preparation procedure, the saturation moments of cubic-magnetic MOFs and triangular cone-magnetic MOFs increased slightly. The magnetic MOFs were predicted to respond excellently to magnetic fields, making the magnetic MOFs and water samples separate easily.

#### 2.1.3. XPS Analysis of NH_2_-Functionalized Nano-Sized Magnetic MOFs

The chemical composition of NH_2_-functionalized nano-sized magnetic MOFs were studied by XPS. According to the results shown in Figure 4, characteristic signals for oxygen (O 1s) and carbon (C 1s) were directly found at 532.2 eV and 284.6 eV. The appearance of a signal at 398.8 eV was assigned to nitrogen (N 1s), which was introduced by the functional group of EDA. Furthermore, another interesting phenomenon found was that with an increasing amount of ammonium hydroxide used in the preparation procedure, the abundance of nitrogen increased remarkably for the magnetic MOFs. The N 1s high-resolution scan of spherical-magnetic MOFs, cubic-magnetic MOFs and triangular cone-magnetic MOFs are shown in Figure 5. The dependence of the abundance of nitrogen in magnetic MOFs on the usage amount of ammonium hydroxide may be explained by the fact that with the increasing ratio of ammonium hydroxide in the reaction system, the alkaline was increased, and amino groups of EDA chelate more easily with Fe, resulting in a higher chance of forming magnetic MOFs.

### 2.2. Effect of pH Value on the Adsorption Efficiencies

To investigate the effect of the pH of the water sample on the adsorption efficiencies of the magnetic MOFs, 40 mL of spiked water sample (50 mg/L, Cr(VI)) with pH values ranging from 2.0 to 9.0 were adsorbed with 50 mg magnetic MOFs. As shown in Figure 6, adsorption efficiencies of magnetic MOFs were remarkably pH dependent. With the water pH value increased from 2.0 to 9.0, the percentage of adsorption of Cr(VI) extremely decreased from 99.07% to 3.51%, 99.23% to 5.76% and 99.91% to 6.40% for spherical-magnetic MOFs, cubic-magnetic MOFs and triangular cone-magnetic MOFs, respectively. The dependence of the adsorption properties of magnetic MOFs to Cr(VI) on sample pH may be illustrated by ion exchange and electrostatic attraction in the adsorption process of Cr(VI) from the water sample by magnetic MOFs. The surfaces of magnetic MOFs are covered with -NH- groups, which can be varied in form with different pH values. Under lower pH values (pH ranging from 2.0 to 6.5), Cr(VI) mainly exists in the soluble form of HCrO_4_^−^; meanwhile, -NH- groups are easily protonated, and so, the electrostatic attraction occurs as shown in Equation (1),
-NH_2_-^+^ + HCrO_4_^−^ → -NH_2_-^+^⋯⋯HCrO_4_^−^
(1)

Furthermore, -NH_2_-^+^ can also contact Cl^−^, and ion exchange will happen between HCrO_4_^−^ and Cl^−^ as in Equation (2),
-NH_2_-^+^Cl^−^ + HCrO_4_^−^ → -NH_2_-^+^HCrO_4_^−^ + Cl^−^
(2)

When the pH value increased, the concentration of H^+^ in the solution decreased, resulting in a lower chance for -NH- group protonation; however, the concentration of OH^−^ concurrently increased, which can be competed with HCrO_4_^−^. As a result, the adsorption ability of magnetic MOFs to Cr(VI) was reduced, resulting in a decreased adsorption efficiency.

### 2.3. Effect of Initial Concentration of Cr(VI) on the Adsorption Efficiencies

The effect of initial concentration on the adsorption of Cr(VI) was studied for magnetic MOFs by varying the initial concentration of Cr(VI) from 50 mg/L to 1000 mg/L. According to the results in Figure 7, it can be seen that the adsorption efficiencies of spherical-magnetic MOFs onto Cr(VI) decreased with the increasing initial concentration of Cr(VI); however, the adsorption capacities of spherical-magnetic MOFs onto Cr(VI) increased. In the case of cubic-magnetic MOFs and triangular cone-magnetic MOFs, similar curve shapes were also found. This phenomenon can be explained by the fact that, in the case of the fixed dosage of magnetic MOFs, the usable adsorption sites are finite, resulting in a clear decrease in the removal of Cr(VI) with increasing initial concentration. Meanwhile, the adsorption capacity of magnetic MOFs to Cr(VI) increased before it obtained the maximum adsorption capacities with increasing initial concentration. In this work, the equilibrium concentration for spherical-magnetic MOFs, cubic-magnetic MOFs and triangular cone-magnetic MOFs were 450, 550 and 600 mg/L, while the adsorption capacities were 198.42, 223.18 and 263.50 mg/g, respectively (Figure 8).

### 2.4. Effect of Usage Amount of Ammonium Hydroxide

As mentioned above, it can be concluded that the adsorption properties of magnetic MOFs were obviously affected by preparation techniques, and so the effect of the usage amount of ammonium hydroxide in the preparation process on the adsorption capacities of magnetic MOFs was investigated. Here, the Langmuir adsorption isotherm was used for the determination of the maximum adsorption capacity of the magnetic MOFs. The adsorption isotherms of magnetic MOFs were obtained by varying the initial concentration of Cr(VI) from 50 to 1000 mg/L with pH 2.0. According to the results in Figure 9, it is clearly seen that the adsorption data can be described effectively by a Langmuir model with R^2^ > 0.999. Therefore, the applicability of Cr(VI) monolayer adsorption onto magnetic MOFs was confirmed. The Langmuir isotherm and constants of magnetic MOFs of adsorption data to Langmuir equations are shown in Table 1. By comparison of the *q_m_* of Cr(VI) onto magnetic MOFs, it increased from 204.08 mg/g to 270.27 mg/g with an increased amount of ammonium hydroxide from 0 to 5 mL used for magnetic MOF preparation. The dependence of the *q_m_* of Cr(VI) on the usage amount of ammonium hydroxide may be related to the morphology of spherical-magnetic MOFs, cubic-magnetic MOFs and triangular cone-magnetic MOFs. Furthermore, the results were also consistent with the findings in the XPS analysis above. The values of *q_m_* increased from 204.08 mg/g to 270.27 mg/g, with an increasing nitrogen percentage for spherical-magnetic MOFs, cubic-magnetic MOFs and triangular cone-magnetic MOFs, which indicated that -NH- groups played a key role in the adsorption process.

### 2.5. Kinetic Studies

To investigate the adsorption kinetic process of Cr(VI) onto magnetic MOFs, batch experiments were obtained by varying the adsorption time from 1.0 to 90 min at a temperature of 35 °C with pH at 2.0; the results are shown in Figure 10. According to the results, the adsorption rate of magnetic MOFs to Cr(VI) was initially very high and slowly achieved an equilibrium in 30 min. Although it took 30 min to achieve equilibrium, the adsorption capacities of magnetic MOFs to Cr(VI) obtained at any time (*q_t_*) got to ~97% of that at equilibrium within the first 5 min. The adsorption kinetic data have been analyzed by a pseudo-second-order model, and related parameters such as rate constant (*k*_2_) and calculating equilibrium adsorption capacity (*q_e_._c_*) are listed in Table 2, which indicated that the adsorption kinetic process of Cr(VI) onto magnetic MOFs well fit the pseudo-second-order rate equation. The *q_e_._c_* obtained from the pseudo-second-order model (37.87, 38.76 and 39.84 mg/g) were close to the experimental *q_t_* (37.85, 38.79 and 39.96 mg/g) for spherical-magnetic MOFs, cubic-magnetic MOFs and triangular cone-magnetic MOFs, respectively. It indicated that the pseudo-second-order model can be used for the prediction of adsorption kinetics. The adsorption capacities of magnetic MOFs were proportional to the active sites on the surface of adsorbents [52]. Along with increased -NH- groups, active sites on the surface of magnetic MOFs increased as well, which resulted in the increase of *q_t_* for the adsorbents.

### 2.6. Adsorption Properties Comparison

The adsorption properties of magnetic MOFs for Cr(VI) removal have been compared with other adsorbents reported in the literature. As shown in Table 3, magnetic MOFs had much higher maximum adsorption capacities compared with most other adsorbents, and triangular cone-magnetic MOFs had the best overall performance for the removal of Cr(VI). Therefore, magnetic MOFs are promising adsorbents for Cr(VI) removal in wastewater.

## 3. Experimental

### 3.1. Materials and Physical Measurements

All the analytical grade reagents were purchased from commercial source, and the dilutions were prepared by using ultrapure water obtained from Milli-Q ultra-pure water system (Millipore, Bedford, MA, USA) with resistivity of 18.2 MΩ.cm. Morphological characteristics of the NH_2_-functionalized nano-size magnetic MOFs were carried out on a Zeiss Supra 55 Field emission scanning electron microscope (SEM) (Zeiss, Oberkochen, Germany). Magnetic properties were performed on a Lake Shore 7404 vibrating sample magnetometer (VSM) (Lake Shore, Columbus, OH, USA). The structures were determined by Ultima IV polycrystal (powder) X-ray diffractometer (XRD) (Rigaku Co., Ltd., Tokyo, Japan), and the instrument was equipped with a copper anode generating Cu Kα radiation (λ = 1.5406 Å). XPS data were collected by ESCALab220i-XL electron spectrometer from VG Scientific (VG Scientific, St. Leonards, UK). Determination of Cr(VI) after adsorption was carried out on a flame atomic absorption spectroscopy (FAAS) (Shimadzu AA-6701F, Shimadzu, Tokyo, Japan), and all of the parameters for FAAS are listed in Table 4.

### 3.2. Synthesis of NH_2_-Functionalized Nano-Sized Magnetic MOFs

The NH_2_-functionalized nano-size magnetic MOFs were synthesized by solvothermal method. Briefly, 5.0 g FeCl_3_·6H_2_O was added to 60 mL ethylene glycol, and then 8.0 g sodium acetate and 10 mL EDA were added under ultrasonication at 60 °C to give a transparent solution. Finally, the mixture was transferred to a 100 mL teflon-lined stainless steel autoclave and continuously reacted at 200 °C for 8 h. After reaction, the autoclave naturally cooled to room temperature. The brown spherical-magnetic MOF nanoparticles were obtained and washed with ethanol and water under ultrasonic conditions. The obtained spherical-magnetic MOFs were dried and stored for further use. The schematic process of NH_2_-functionalized nano-sized magnetic MOF synthesis is shown in Figure 11.

The other two kinds of NH_2_-functionalized nano-sized magnetic MOFs, i.e., cubic-magnetic MOFs and triangular cone-magnetic MOFs, were synthesized in a similar way by adding 1.0 mL and 5.0 mL ammonium hydroxide, respectively.

### 3.3. Adsorption Experiments

The stock Cr(VI) solution (1000 mg/L) was prepared by dissolving K_2_Cr_2_O_7_ in water. Batch adsorption experiments were carried out by mixing 50 mg NH_2_-functionalized nano-sized magnetic MOFs with 40 mL Cr(VI) solution of varying concentrations from 10 to 1000 mg/L under continuous stirring. NaOH (0.5 mol/L) and HCl (1.0 mol/L) solutions were applied to pH adjustment. To investigate the effect of pH on the adsorption efficiency of NH_2_-functionalized nano-sized magnetic MOFs, batch adsorption experiments were carried out with pH ranging from 2.0 to 9.0 for 24 h to obtain equilibrium. Additionally, the effect of initial Cr(VI) concentrations and the different morphology of NH_2_-functionalized nano-sized magnetic MOFs were investigated by varying initial Cr(VI) concentrations from 10 to 1000 mg/L with pH 2.5. The adsorption kinetic experiments were investigated with adsorption time ranging from 1 to 120 min, and the Cr(VI) concentrations were measured at given time intervals by FAAS.

### 3.4. Analysis of Adsorption Data

#### 3.4.1. Adsorption Model

Adsorption capacities *q_e_* (mg/g) and removal efficiency were determined by using the equation below:(3)$$q_{e}=\frac{\left(C_{0}-C_{e}\right)V}{m}$$
where *C*_0_ is initial Cr(VI) concentration, and *C_e_* is equilibrium Cr(VI) concentration (mg/L), *m* is the dry weight of NH_2_-functionalized nano-size magnetic MOFs (mg), and *V* is volume of Cr(VI) solutions (mL), respectively.

Here, the Langmuir adsorption isotherm was used for the determination of maximum adsorption capacity of NH_2_-functionalized nano-sized magnetic MOFs. Because the adsorption data on the adsorption of Cr(VI) to the NH_2_-functionalized nano-sized magnetic MOFs were found to match the Langmuir model, which is expressed below [55]:(4)$$\frac{C_{e}}{q_{e}}=\frac{1}{Kq_{m}}+\frac{C_{e}}{q_{m}}$$

In the Langmuir equation, *q_m_* is expressed as the maximum adsorption capacity, and *K* is Langmuir constant related to apparent heat change.

#### 3.4.2. Kinetic Model

In this work, the pseudo-second-order model has been used for the evaluation adsorption kinetic data on the adsorption of Cr(VI) to the NH_2_-functionalized nano-sized magnetic MOFs, and the equation is expressed as below [55]:(5)$$\frac{t}{q_{t}}=\frac{1}{{k_{2}q_{e.c}}^{2}}+\frac{t}{q_{e.c}}$$

In the pseudo-second-order equation, *q_t_* is the amount of Cr(VI) adsorbed onto NH_2_-functionalized nano-sized magnetic MOFs any time *t* (mg/g) and *k*_2_ is second-order rate constant at equilibrium ((g/mg)/min).

## 4. Conclusions

In this work, NH_2_-functionalized nano-sized magnetic metal–organic frameworks, i.e., spherical-magnetic MOFs, cubic-magnetic MOFs and triangular cone-magnetic MOFs, were prepared by the solvothermal method by varying the usage amount of ammonium hydroxide during the preparation process. The adsorption effectiveness of the magnetic MOFs for Cr(VI) removal in wastewater was verified by laboratory batch tests. The adsorption efficiency was highly pH dependent, and the Cr(VI) adsorption process was consistent with the Langmuir isotherm model, revealing that the adsorption process was chemical and molecular layer adsorption. The maximum adsorption capacity of spherical-magnetic MOFs, cubic-magnetic MOFs and triangular cone-magnetic MOFs was 204.08 mg/g, 232.56 mg/g and 270.27 mg/g, respectively, which is much higher than the adsorption capacities of other adsorbents reported in most of the previous studies.

## Figures and Tables

**Figure 1 molecules-29-01007-f001:**
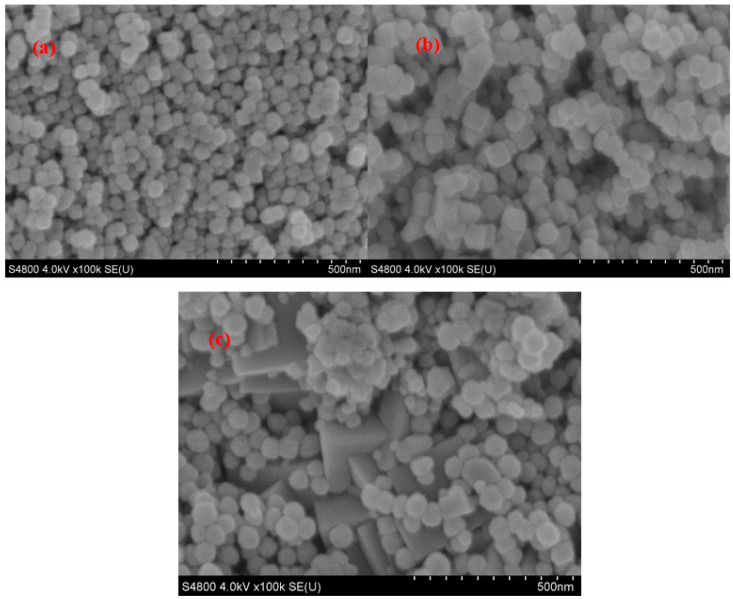
Scanning electron microscopy (SEM) images of the NH_2_-functionalized nano-sized magnetic metal–organic frameworks (MOFs), i.e., spherical-magnetic MOFs (**a**), cubic-magnetic MOFs (**b**) and triangular cone-magnetic MOFs (**c**).

**Figure 2 molecules-29-01007-f002:**
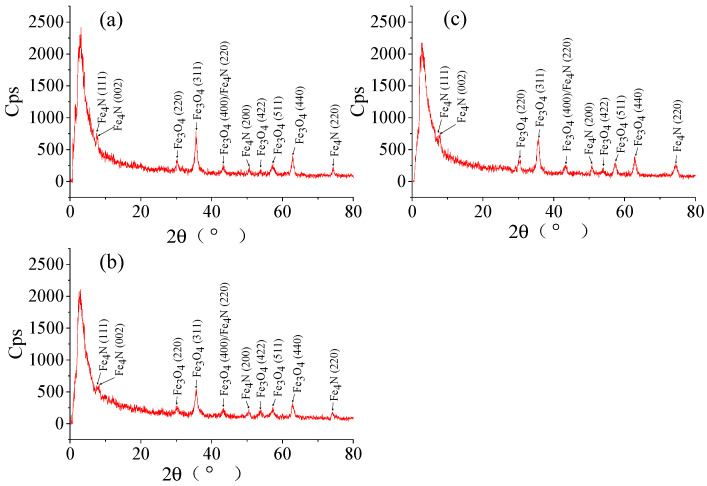
X-ray diffraction (XRD) patterns of three kinds of magnetic metal–organic frameworks (MOFs), i.e., spherical-magnetic MOFs (**a**), cubic-magnetic MOFs (**b**) and triangular cone-magnetic MOFs (**c**).

**Figure 3 molecules-29-01007-f003:**
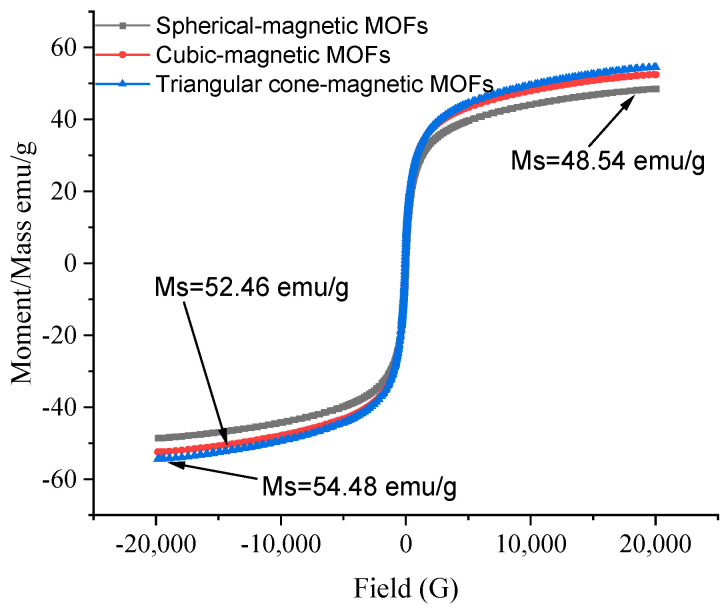
Vibrating sample magnetometer (VSM) analysis of NH_2_-functionalized nano-size magnetic metal–organic frameworks (MOFs).

**Figure 4 molecules-29-01007-f004:**
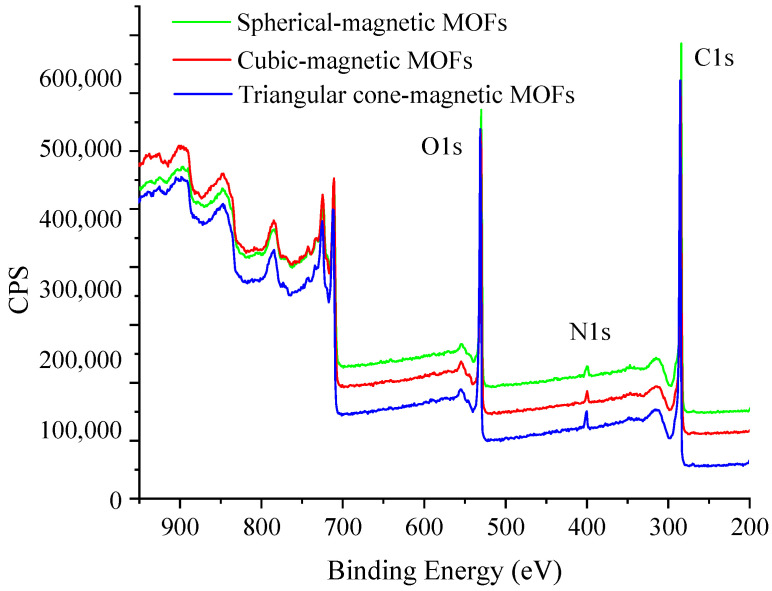
X-ray photoelectron spectroscopy (XPS) analysis of the magnetic metal–organic frameworks (MOFs).

**Figure 5 molecules-29-01007-f005:**
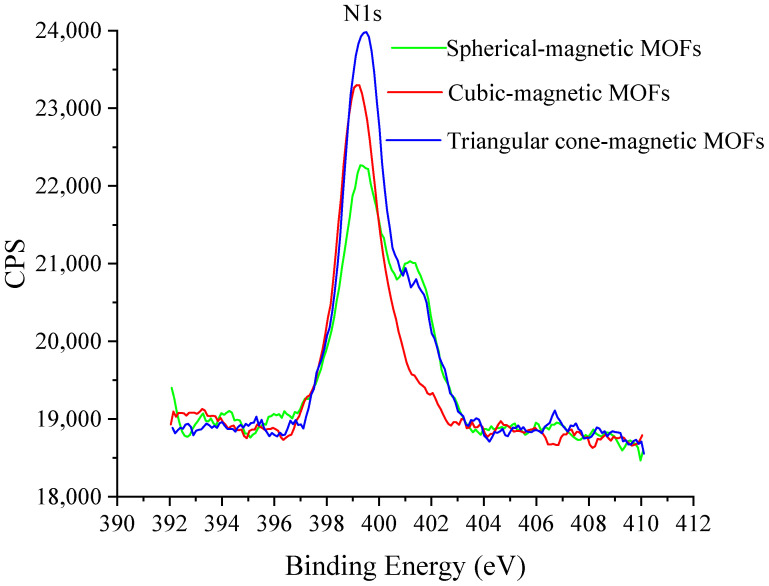
X-ray photoelectron spectroscopy (XPS) analysis of the N1s high-resolution scan of the magnetic metal–organic frameworks (MOFs).

**Figure 6 molecules-29-01007-f006:**
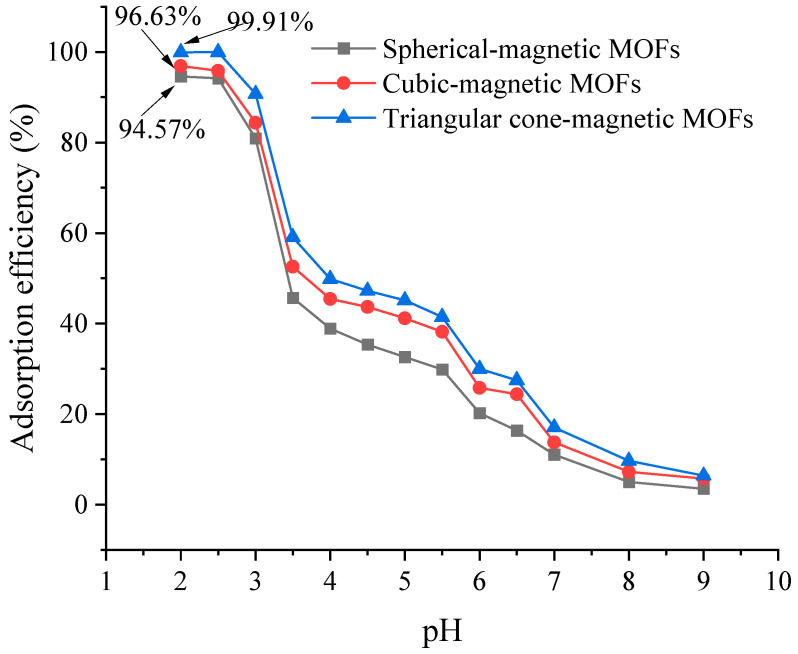
Effect of pH value on the adsorption efficiencies.

**Figure 7 molecules-29-01007-f007:**
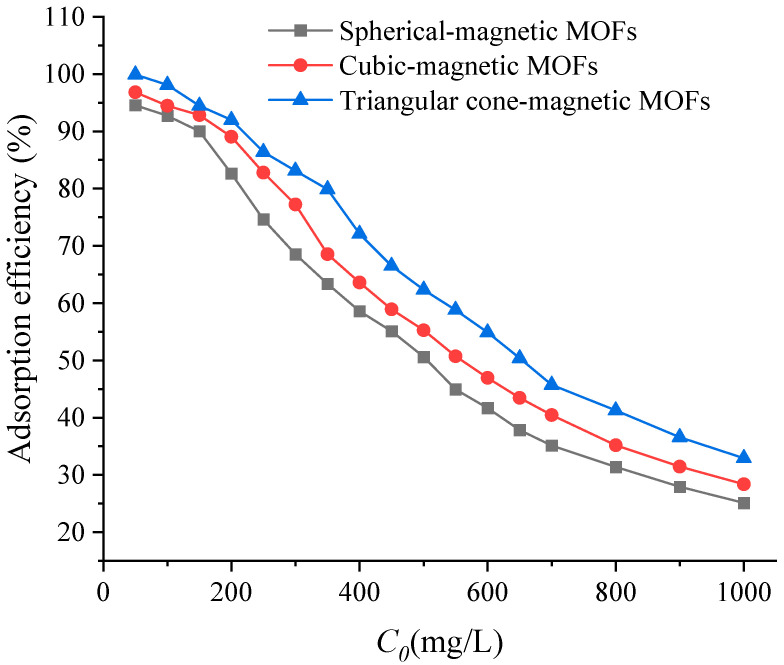
Effect of initial concentration of Cr(VI) on the adsorption efficiencies.

**Figure 8 molecules-29-01007-f008:**
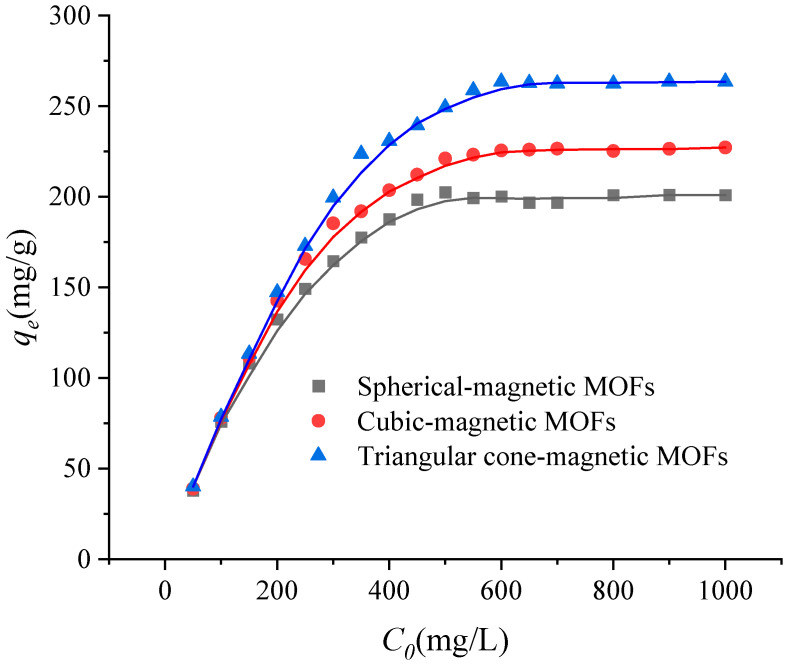
Effect of initial concentration of Cr(VI) on the adsorption capacities.

**Figure 9 molecules-29-01007-f009:**
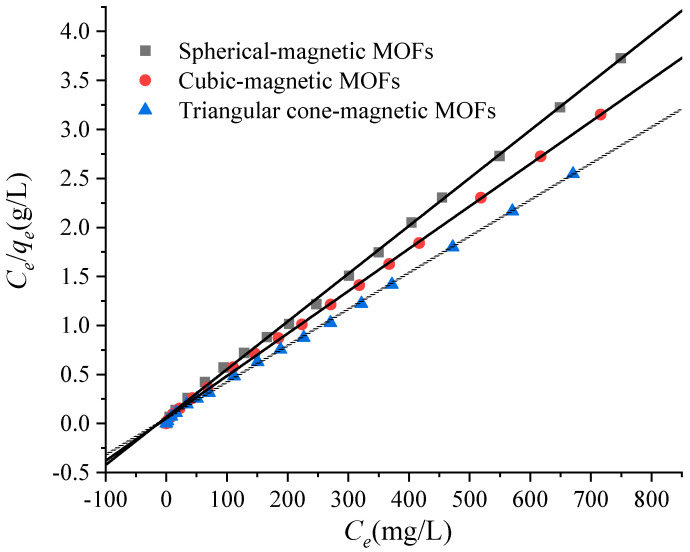
Langmuir isotherms for the adsorption of Cr(VI) onto magnetic metal-organic frameworks (MOFs).

**Figure 10 molecules-29-01007-f010:**
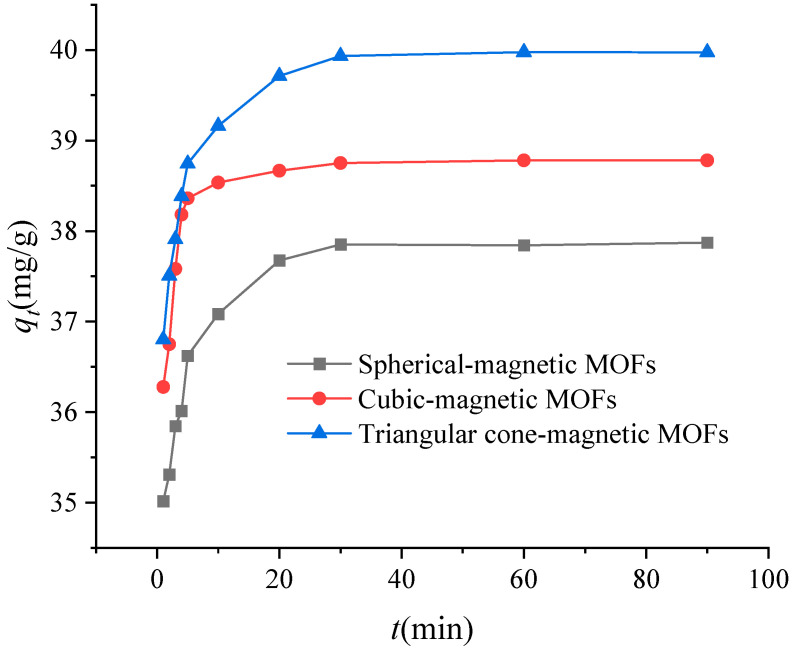
Kinetic studies on the adsorption of Cr(VI) by magnetic metal–organic frameworks (MOFs).

**Figure 11 molecules-29-01007-f011:**
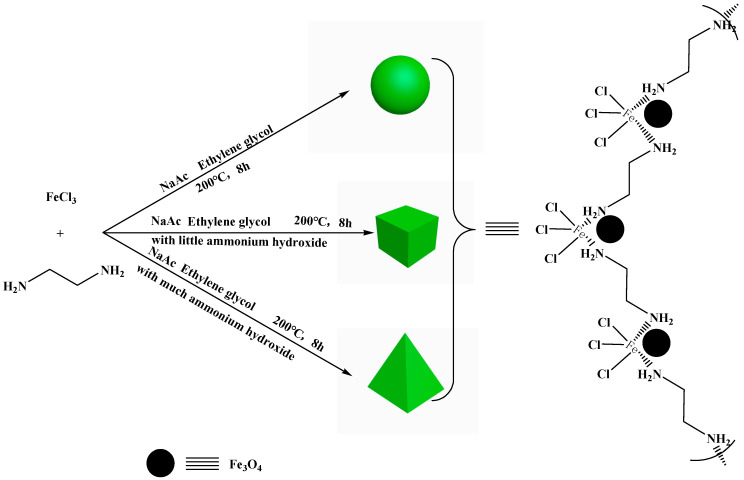
The schematic procedure of NH_2_-functionalized nano-size magnetic metal–organic framework (MOF) synthesis.

**Table 1 molecules-29-01007-t001:** The Langmuir isotherms and constants of magnetic metal–organic frameworks (MOFs).

Magnetic MOFs	Langmuir Isotherms	Langmuir Constants		
		*K* (L/mg)	*R* ^2^	*q_m_* (mg/g)
Spherical-magnetic MOFs	*C_e_*/*q_e_* = 0.0049*C_e_* + 0.0652	0.0752	0.9992	204.08
Cubic-magnetic MOFs	*C_e_*/*q_e_* = 0.0043*C_e_* + 0.0531	0.0810	0.9994	232.56
Triangular cone-magnetic MOFs	*C_e_*/*q_e_* = 0.0037*C_e_* + 0.0443	0.0835	0.9994	270.27

**Table 2 molecules-29-01007-t002:** The pseudo-second-order rate equations and constants of magnetic metal–organic frameworks (MOFs).

Magnetic MOFs	Pseudo-Second-Order Rate Equations	*k*_2_ [(g/mg)/min]	*q_t_* (mg/g)	*q_e_._c_* (mg/g)	*k*_2_*q_e_._c_*^2^ [(mg/g)/min]	*R* ^2^
Spherical-magnetic MOFs	*t*/*q_t_* = 0.0264*t* + 0.0042	0.1660	37.85	37.87	238.1	1
Cubic-magnetic MOFs	*t*/*q_t_* = 0.0258*t* + 0.0040	0.1664	38.79	38.76	250.0	1
Triangular cone-magnetic MOFs	*t*/*q_t_* = 0.0251*t* + 0.0038	0.1658	39.96	39.84	263.2	1

**Table 3 molecules-29-01007-t003:** Adsorption capacities of various adsorbents for Cr(VI).

Adsorbents	Equilibrium Time (min)	pH	*q_m_* (mg/g)	Ref.
water-stable Cu(II)-MOF material	180	6.0	190	[50]
EDA-MPs	5~60	2.5	32.15~61.35	[55]
nZVI/Kaol	10	<4.0	15.98	[57]
mZVI/GCS	288 h	2.5	243.63	[58]
Fe-TA-PEI-10	120	3.0	161.6	[59]
Mo-Fe0.5-CS	30	-	180	[60]
MOF-DFSA	120	4.0	188.12	[61]
Fe@N-L-GM	300	2.0	65.83	[62]
CsSB/MnFe_2_O_4_@BC	360	2.0	125.34	[63]
N–CHC	240	2.0	151.05	[64]
Spherical-magnetic MOFs	30	2.0	204.08	This work
Cubic-magnetic MOFs	30	2.0	232.56	This work
Triangular cone-magnetic MOFs	30	2.0	270.27	This work

nZVI/Kaol: Kaolinite-based nano zero-valent iron composite mZVI/GCS: glutaraldehyde-crosslinked chitosan-encapsulating microscale zero-valent iron Fe-TA-PEI-10: tannic acid and polyethyleneimine-modified zero-valent iron particles Mo-Fe0.5-CS: Chitosan-induced synthesis of few-layer MoS_2_/Fe-doped biochar Fe@N-L-GM: aminated lignin/geopolymer supported with Fe nanoparticles CsSB/MnFe_2_O_4_@BC: modified biochar with chitosan Schiff base and MnFe_2_O_4_ nanoparticles N–CHC: N-doped cellulose-based hydrothermal carbon.

**Table 4 molecules-29-01007-t004:** Instrumental conditions for Cr(VI) determination and parameters of flame atomic absorption spectroscopy (FAAS) method.

Spectrometer Parameter	
(a)	
Wavelength	357.9 nm
Slit width	0.2 nm
Lamp current	8 mA
Burnet height	9 mm
Background correction	Deuterium lamp
Parameters	
(b)	
Linearity range	50–1250 μg·L^−1^
Detection limit	20 μg·L^−1^

## Data Availability

Details are available from the authors.

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
