# Peer review of "Orientation Growth of N-Doped and Iron-Based Metal–Organic Framework and Its Application for Removal of Cr(VI) in Wastewater"

_molecules, 2024, doi:10.3390/molecules29051007_

Round 1
Reviewer 1 Report (Previous Reviewer 1)
Comments and Suggestions for Authors
Grammar check may be necessary. Typos can be found, e.g. "con" at line 158.
Better quality figures are needed, especially figure 3.
Comments on the Quality of English LanguageTypos should be corrected, and reference list improved, e.g. journal name abbreviations, for refs 23,24.
Author Response
Dear reviewer,
We are truly grateful to your critical comments and thoughtful suggestions. Based on these comments and suggestions, we have made careful modifications on the original manuscript. All changes made to the text are in red color. We hope the new manuscript will meet the magazine’s standard. Below you will find our point-by-point responses to your comments/questions:
Comments and Suggestions for Authors:
Grammar check may be necessary. Typos can be found, e.g. "con" at line 158.
Better quality figures are needed, especially figure 3.
√ Many thanks for your suggestion, we are very sorry for the mistake, the “triangular con” has been corrected to “triangular cone”. Furthermore, we have checked the manuscript carefully, some mistakes have been corrected. And new version of figure 3 has been added to the manuscript, thank you very much!
Figure 3. The XRD patterns of three kinds of magnetic MOFs, i.e., spherical-magnetic MOFs (a), cube-magnetic MOFs (b) and triangular cone-magnetic MOFs(c).
Comments on the Quality of English Language:
Typos should be corrected, and reference list improved, e.g. journal name abbreviations, for refs 23,24.
√ Many thanks for your suggestion, and we greatly appreciate your nice comments on our article. According to your suggestion, we have checked the references carefully, and some typos have been corrected. Thank you very much!
Yours sincerely
Prof. Yong-Gang Zhao
College of Biological and Environmental Engineering, Zhejiang Shuren University, Hangzhou, 310015, China
E-mail address: zhyg91213@163.com

Reviewer 2 Report (Previous Reviewer 2)
Comments and Suggestions for Authors
The paper is significantly improved compared to the previously submitted form. All the comments of the reviewers were adopted, and a detailed XRD analysis was also introduced, which significantly contributes to the value of the work. Therefore I suggest to be accepted for publication.
Author Response
Comments and Suggestions for Authors:
The paper is significantly improved compared to the previously submitted form. All the comments of the reviewers were adopted, and a detailed XRD analysis was also introduced, which significantly contributes to the value of the work. Therefore, I suggest to be accepted for publication.
√ Many thanks for your suggestion, and we greatly appreciate your nice comments on our article.
Yours sincerely
Prof. Yong-Gang Zhao
College of Biological and Environmental Engineering, Zhejiang Shuren University, Hangzhou, 310015, China
E-mail address: zhyg91213@163.com
This manuscript is a resubmission of an earlier submission. The following is a list of the peer review reports and author responses from that submission.
Round 1
Reviewer 1 Report
Comments and Suggestions for Authors
Authors prepared materials claimed to be magnetic MOFs, and studied their performance for adsorption of Cr(IV). The morphology was revealed by SEM and X-ray diffraction, and magnetic properties measured using vibration sample magnetometry. The results may be interesting for future applications, however there are major problems with the study, which requires major revision.
Major corrections
1. Authors must prove that the obtained materials were indeed MOFs. Higher accuracy X-ray diffraction should be done, and characteristic strong reflections at appropriate angles should be obtained. The present patterns indicate only the formation of iron oxide phase.
2. The origin of the recipe used for the MOF synthesis should be described, and references supplied.
3. In the conclusions, author write that the sorption was following Freundlich model. However, this is not shown in the manuscript. Please add also the corresponding analysis.
4. In the introduction, several more recent works can be mentioned with nanomaterials featuring adsorption of Chromium, and related to the present study: amino-functionalized silica (Materials 2021, 14, 628) , amino-functionalized magnetic polymers (Materials 2023, 16, 2233) and metallo organic frameworks MOFs, (Adv. Mater. Interfaces 2023, 2300424)
Minor corrections
1. The axes notations in the graphs should be improved, following the usual examples from the literature. Thus, x-axes of figs 3,4, and unifying the spelling of the units in figs 5,6.
Author Response
Dear editor,
We are truly grateful to yours and reviewers’ critical comments and thoughtful suggestions. Based on these comments and suggestions, we have made careful modifications on the original manuscript. All changes made to the text are in red color. We hope the new manuscript will meet your magazine’s standard. Below you will find our point-by-point responses to the reviewers’ comments/questions:
Authors prepared materials claimed to be magnetic MOFs, and studied their performance for adsorption of Cr(IV). The morphology was revealed by SEM and X-ray diffraction, and magnetic properties measured using vibration sample magnetometry. The results may be interesting for future applications, however there are major problems with the study, which requires major revision.
√ Many thanks for your suggestion. We greatly appreciate your kind suggestion.
Major corrections
- Authors must prove that the obtained materials were indeed MOFs. Higher accuracy X-ray diffraction should be done, and characteristic strong reflections at appropriate angles should be obtained. The present patterns indicate only the formation of iron oxide phase.
√ Many thanks for your suggestion. We greatly appreciate your kind thoughts. In this work, a series of NH2-functionalized nano-size magnetic metal organic framework (MOFs) based on the nano-size magnetic Fe3O4 was prepared. In these MOFs, the magnetic Fe3O4 cores were covered with Fe-N groups, which can be seen in the X-ray diffraction patterns. Beside the six characteristic peaks of Fe3O4 at 2θ of 30.1o, 35.5 o, 43.1 o, 53.4 o, 57.0 o and 62.6 o found in the XRD patterns of magnetic MOFs, which corresponded to their indices (220), (311), (400), (422), (511) and (400). Furthermore, peaks of Fe4N at 2θ of 41.3 o, 47.5 o and 74.6 o were found in the XRD patterns of magnetic MOFs, which corresponded to their indices (111), (200) and (220).
Moreover, in order to further prove that the obtained materials were indeed MOFs. Higher accuracy X-ray diffraction will be carried out in our further work. And we will keep this in mind in the further study. Thank you very much!
- The origin of the recipe used for the MOF synthesis should be described, and references supplied.
√ Many thanks for your suggestion. We greatly appreciate your kind thoughts. And the origin of the recipe used for the MOF synthesis has been added to the “Introduction” as below, “Among these adsorption materials, MOFs is a kind of crystalline porous material with periodic network structure formed by the interconnection of inorganic metal centers (metal ions or metal clusters) and bridging organic ligands through self-assembly [43-45]. As novel organic-inorganic hybrid materials, MOFs is different from inorganic porous materials and general organic complexes. It combines the rigidity of inorganic materials with the flexibility of organic materials.”
Thank you very much!
- In the conclusions, author write that the sorption was following Freundlich model. However, this is not shown in the manuscript. Please add also the corresponding analysis.
√ Many thanks for your suggestion. We greatly appreciate your kind thoughts. In this work, the Langmuir model and Freundlich model were both used to characterize the maximum adsorption capacity of the given adsorbents, the results showed that the adsorption data for Cr(VI) onto magnetic MOFs were well fitted with the Langmuir isotherm, but not well for the Freundlich model. And so, the Freundlich model was not listed in the manuscript. In the conclusions, we are very sorry for the mistake, and “Freundlich model” has been deleted. Thank you very much!
- In the introduction, several more recent works can be mentioned with nanomaterials featuring adsorption of Chromium, and related to the present study: amino-functionalized silica (Materials 2021, 14, 628), amino-functionalized magnetic polymers (Materials 2023, 16, 2233) and metallo organic frameworks MOFs, (Adv. Mater. Interfaces 2023, 2300424)
√ Many thanks for your suggestion. We greatly appreciate your kind thoughts. And the related references have been cited in the work as below, thank you very much!
- Putz, A.-M.; Ciopec, M.; Negrea, A.; Grad, O.; Ianăşi, C.; Ivankov, O.I.; Milanović, M.; Stijepović, I.; Almásy, L. Comparison of structure and adsorption properties of mesoporous silica functionalized with aminopropyl groups by the co-condensation and the post grafting methods. Materials 2021, 14, 628
- Suručić, L.; Janjić, G.; Marković, B.; Tadić, T.; Vuković, Z.; Nastasović, A.; Onjia, A. Speciation of hexavalent chromium in aqueous solutions using a magnetic silica-coated amino-modified glycidyl methacrylate polymer nanocomposite. Materials 2023, 16, 2233
- Valverde, A.; Luis, R. de F.-de; Salazar, H.; Gonçalves, B.F.; King, S.; Almásy, L.; Kriechbaum, M.; Laza, J.M.; Vilas-Vilela, J.L.; Martins, P.M.; Lanceros-Mendez, S.; Porro, J.M.; Petrenko, V.I. On the multiscale structure and morphology of PVDF-HFP@MOF membranes in the scope of water remediation applications. Adv. Mater. Interfaces 2023, 2300424.
Minor corrections
- The axes notations in the graphs should be improved, following the usual examples from the literature. Thus, x-axes of figs 3,4, and unifying the spelling of the units in figs 5,6.
√ Many thanks for your suggestion. We greatly appreciate your kind thoughts, and we are very sorry for the mistakes. They have been corrected, thank you very much!

Reviewer 2 Report
Comments and Suggestions for Authors
The present study, although with an interesting topic, is incomprehensible to readers. I suggest a major revision before publishing.
Throughout the whole paper the authors refer to used materials as NH2-functionalized nano-size magnetic metal organic framework. How is concluded that it is nano-size material, from SEM (Figure 2) it can be seen that diameter of particle is 500 nm, and in line 149 it is written that MOFs are spherical with diameter of 50 nm?
Line 15: „the amount increasing“ should be replaced with „increasing the amount“
Line 16: „morphological“ should be replaced with „morphology“
Line 18: „could up to“ should be replaced with „could be up to“
Line 31: „if inhalation“ should be replaced with „from inhalation“
Line 32: „long-term exposure of Cr(VI) and so Cr(VI) ions has been“ should be replaced with „long-term exposure to Cr(VI) and so Cr(VI) ions have been“
Line 34: detected in wastewater should be replaced with detected heavy metals in wastewater
Line 35: Therefore, it is essential that study on the Cr(VI) removal in water to solve heavy metal pollution should be replaced with Therefore, it is essential to study the Cr(VI) removal from water to solve heavy metal pollution
Line 37: much more should be replaced with many
Line 42: technology should be replaced with technique
Line 55: adsorption should be replaced with adsorbs
Line 59: been should be deleted
Line 67: combine should be replaced with by combining
Line 68: would be probably become should be replaced with would probably become
Line 72: infra should be deleted
Line 73: micrographs should be replaced with microscopy
The last paragraph in Introduction states characterization techniques: SEM, VSM and XPS.
Why XRD is not mentioned here, since it is described later in the Results?
Line 82: micrographs should be replaced with microscope
Line 88: were should be replaced with was
Line 114: morphological should be replaced with morphology
Line 124: concentrations should be replaced with concentration
Line 140: amounts should be replaced with amount
Line 148: morphological should be replaced with morphologies
Line 150: the usage amount of ammonium hydroxide increased in the preparation procedure, the morphological of magnetic MOFs should be replaced with an increased amount of ammonium hydroxide used in the preparation procedure, the morphology of magnetic MOFs
Line 174: is lower, while with the usage amount of ammonium hydroxide increased in the preparation procedure should be replaced with are low, while with increasing the amount of ammonium hydroxide used in the preparation procedure
Line176: is should be deleted
Line 187: the usage amount of ammonium hydroxide increased in the preparation procedure
should be replaced with increasing the amount of ammonium hydroxide used in the preparation procedure
Line 193: and amino groups of EDA was much easier to be chelated with Fe resulting in the more chance of formation magnetic MOFs should be replaced with and amino groups of EDA chelate more easily with Fe resulting in a higher chance of forming magnetic MOFs
Line 203: 50 g? Is this correct? I think it should be 50 mg.
Line 257: morphological should be replaced with morphology
Line 271: to an equilibrium in 30 min should be replaced with an equilibrium in 30 min
Line 283: also should be deleted
In the conclusion it is stated:
„the Cr(VI) adsorption process was consistent with both Langmuir and Freundlich isotherms models, revealing the adsorption process was chemical and molecular layer adsorption.“
How is that possible since this is the first time that Freundlich isotherms are mentioned in this paper?
Comments on the Quality of English LanguageThe english is very bad, and often the reader has to guess what the authors wanted to say.
Author Response
Dear editor,
We are truly grateful to yours and reviewers’ critical comments and thoughtful suggestions. Based on these comments and suggestions, we have made careful modifications on the original manuscript. All changes made to the text are in red color. We hope the new manuscript will meet your magazine’s standard. Below you will find our point-by-point responses to the reviewers’ comments/questions:
The present study, although with an interesting topic, is incomprehensible to readers. I suggest a major revision before publishing. Throughout the whole paper the authors refer to used materials as NH2-functionalized nano-size magnetic metal organic framework. How is concluded that it is nano-size material, from SEM (Figure 2) it can be seen that diameter of particle is 500 nm, and in line 149 it is written that MOFs are spherical with diameter of 50 nm?
√ Many thanks for your suggestion. In Figure 2, “500 nm” listed in the SEM spectra is a ruler of measuring the particles, and according to the ruler, MOFs are spherical with diameter of 50 nm. Thank you very much!
Line 15: “the amount increasing” should be replaced with “increasing the amount”
√ Many thanks for your suggestion. We greatly appreciate your kind thoughts. We have corrected the mistake.
Line 16: “morphological” should be replaced with “morphology”
√ Many thanks for your suggestion. We greatly appreciate your kind thoughts. We have corrected the mistake.
Line 18: “could up to” should be replaced with “could be up to”
√ Many thanks for your suggestion. We greatly appreciate your kind thoughts. We have corrected the mistake.
Line 31: “if inhalation” should be replaced with “from inhalation”
√ Many thanks for your suggestion. We greatly appreciate your kind thoughts. We have corrected the mistake.
Line 32: “long-term exposure of Cr(VI) and so Cr(VI) ions has been” should be replaced with “long-term exposure to Cr(VI) and so Cr(VI) ions have been”
√ Many thanks for your suggestion. We greatly appreciate your kind thoughts. We have corrected the mistake.
Line 34: “detected in wastewater” should be replaced with “detected heavy metals in wastewater”
√ Many thanks for your suggestion. We greatly appreciate your kind thoughts. We have corrected the mistake.
Line 35: “Therefore, it is essential that study on the Cr(VI) removal in water to solve heavy metal pollution” should be replaced with “Therefore, it is essential to study the Cr(VI) removal from water to solve heavy metal pollution”
√ Many thanks for your suggestion. We greatly appreciate your kind thoughts. We have corrected the mistake.
Line 37: “much more” should be replaced with “many”
√ Many thanks for your suggestion. We greatly appreciate your kind thoughts. We have corrected the mistake.
Line 42: “technology” should be replaced with “technique”
√ Many thanks for your suggestion. We greatly appreciate your kind thoughts. We have corrected the mistake.
Line 55: “adsorption” should be replaced with “adsorbs”
√ Many thanks for your suggestion. We greatly appreciate your kind thoughts. We have corrected the mistake.
Line 59: “been” should be deleted
√ Many thanks for your suggestion. The “been” has been deleted. Thank you very much!
Line 67: “combine” should be replaced with by “combining”
√ Many thanks for your suggestion. We greatly appreciate your kind thoughts. We have corrected the mistake.
Line 68: “would be probably become” should be replaced with “would probably become”
√ Many thanks for your suggestion. We greatly appreciate your kind thoughts. We have corrected the mistake.
Line 72: “infra” should be deleted
√ Many thanks for your suggestion. We greatly appreciate your kind thoughts, the “infra” has been deleted.
Line 73: “micrographs” should be replaced with “microscopy”
√ Many thanks for your suggestion. We greatly appreciate your kind thoughts. We have corrected the mistake.
The last paragraph in Introduction states characterization techniques: SEM, VSM and XPS. Why XRD is not mentioned here, since it is described later in the Results?
√ Many thanks for your suggestion. We greatly appreciate your kind thoughts. We have added “XRD” to the last paragraph in Introduction. Thank you very much!
Line 82: “micrographs” should be replaced with “microscope”
√ Many thanks for your suggestion. We greatly appreciate your kind thoughts. We have corrected the mistake.
Line 88: “were” should be replaced with “was”
√ Many thanks for your suggestion. We greatly appreciate your kind thoughts. We have corrected the mistake.
Line 114: “morphological” should be replaced with “morphology”
√ Many thanks for your suggestion. We greatly appreciate your kind thoughts. We have corrected the mistake.
Line 124: “concentrations” should be replaced with “concentration”
√ Many thanks for your suggestion. We greatly appreciate your kind thoughts. We have corrected the mistake.
Line 140: “amounts” should be replaced with “amount”
√ Many thanks for your suggestion. We greatly appreciate your kind thoughts. We have corrected the mistake.
Line 148: “morphological” should be replaced with “morphologies”
√ Many thanks for your suggestion. We greatly appreciate your kind thoughts. We have corrected the mistake.
Line 150: “the usage amount of ammonium hydroxide increased in the preparation procedure, the morphological of magnetic MOFs” should be replaced with “an increased amount of ammonium hydroxide used in the preparation procedure, the morphology of magnetic MOFs”
√ Many thanks for your suggestion. We greatly appreciate your kind thoughts. We have corrected the mistake.
Line 174: “is lower, while with the usage amount of ammonium hydroxide increased in the preparation procedure” should be replaced with “are low, while with increasing the amount of ammonium hydroxide used in the preparation procedure”
√ Many thanks for your suggestion. We greatly appreciate your kind thoughts. We have corrected the mistake.
Line176: “is” should be deleted
√ Many thanks for your suggestion. We greatly appreciate your kind thoughts, the “is” has been deleted.
Line 187: “the usage amount of ammonium hydroxide increased in the preparation procedure” should be replaced with “increasing the amount of ammonium hydroxide used in the preparation procedure”
√ Many thanks for your suggestion. We greatly appreciate your kind thoughts. We have corrected the mistake.
Line 193: “and amino groups of EDA was much easier to be chelated with Fe resulting in the more chance of formation magnetic MOFs” should be replaced with “and amino groups of EDA chelate more easily with Fe resulting in a higher chance of forming magnetic MOFs”
√ Many thanks for your suggestion. We greatly appreciate your kind thoughts. We have corrected the mistake.
Line 203: 50 g? Is this correct? I think it should be 50 mg.
√ Many thanks for your suggestion. We greatly appreciate your kind thoughts. It is 50 mg, and we have corrected the mistake. Thank you very much!
Line 257: “morphological” should be replaced with “morphology”
√ Many thanks for your suggestion. We greatly appreciate your kind thoughts. We have corrected the mistake.
Line 271: “to an equilibrium in 30 min” should be replaced with “an equilibrium in 30 min”
√ Many thanks for your suggestion. We greatly appreciate your kind thoughts. We have corrected the mistake.
Line 283: also should be deleted
√ Many thanks for your suggestion. We greatly appreciate your kind thoughts, the “also” has been deleted.
In the conclusion it is stated: “the Cr(VI) adsorption process was consistent with both Langmuir and Freundlich isotherms models, revealing the adsorption process was chemical and molecular layer adsorption.” How is that possible since this is the first time that Freundlich isotherms are mentioned in this paper?
√ Many thanks for your suggestion. We greatly appreciate your kind thoughts. We are very sorry for the mistake, and the “Freundlich isotherms” has been deleted. Thank you very much!

Round 2
Reviewer 1 Report
Comments and Suggestions for Authors
The paper still lacks the proof that the obtained materials were indeed Metal-Organic Frameworks - MOFs. X-ray diffraction must be performed and the strong reflections (111) and (002) at around 7 and 8 degrees (with Cu anode), characteristic for MOFs , should be shown. This can indicate a successful preparation of the title material MOF. Otherwise, the paper cannot be recommended for publication.
Author Response
Dear reviewer,
We are truly grateful to your critical comments and thoughtful suggestions. Based on these comments and suggestions, we have made careful modifications on the original manuscript. All changes made to the text are in red color. We hope the new manuscript will meet the magazine’s standard. Below you will find our point-by-point responses to your comments/questions:
The paper still lacks the proof that the obtained materials were indeed Metal-Organic Frameworks - MOFs. X-ray diffraction must be performed and the strong reflections (111) and (002) at around 7 and 8 degrees (with Cu anode), characteristic for MOFs, should be shown. This can indicate a successful preparation of the title material MOF. Otherwise, the paper cannot be recommended for publication.
√ Many thanks for your suggestion. We greatly appreciate your kind suggestion. Firstly, we are very sorry that the magnetic MOFs were not analyzed by high-resolution XRD.
In this work, the characterization of NH2-functionalized nano-size magnetic MOFs have been tested by third-party testing company (Scientific Compass Test Platform, https://www.shiyanjia.com/). For the RXD analysis, because the magnetic MOFs has strong magnetic properties, and so the analysts suggest that the 2θ would be in the range of 10°~80°.
We also consulted analysts about the XRD results, and the relatively lower reflections of MOF layers may be resulted from the Fe3O4 core occupies the vast majority of magnetic MOFs. And so, the reflections of MOF layers were covered up by Fe3O4 core as shown in the Figure 3 “The XRD patterns of three kinds of magnetic MOFs, i.e., spherical-magnetic MOFs (a), cube-magnetic MOFs (b) and triangular cone-magnetic MOFs(c).”
Secondly, besides the XRD analysis, the chemical composition of magnetic MOFs was studied by XPS. And the N1s high-resolution scan of spherical-magnetic MOFs, cube-magnetic MOFs and triangular cone-magnetic MOFs were shown in Figure 6. The dependence of the abundance of nitrogen in magnetic MOFs on the usage amount of ammonium hydroxide may be explained that with the increasing of the ratio of ammo-nium hydroxide in the reaction system, the alkaline was increased, and amino groups of EDA chelate more easily with Fe resulting in a higher chance of forming magnetic MOFs. And the results were consistent with the RXD analysis.
Figure 6. X-ray photoelectron spectroscopy (XPS) analysis of the N1s high-resolution scan of the magnetic MOFs.
Thirdly, in order to further prove that the obtained materials were indeed MOFs, in the last two weeks, magnetic MOFs has been sent to another third-party testing organization for high-resolution XRD analysis. Till now, the results have not been obtained. And we have been informed that the high-resolution XRD has not been worked properly and it would be fixed in two months. Therefore, the question mentioned above will be studied deeply in our further work. And the results would be listed as supplementary data associated with this work.
Thank you very much!
Yours sincerely
Prof. Yong-Gang Zhao
College of Biological and Environmental Engineering, Zhejiang Shuren University, Hangzhou, 310015, China
E-mail address: zhyg91213@163.com

Reviewer 2 Report
Comments and Suggestions for Authors
If Freundlich isotherm model is deleted from the conclusion, then in line 316 "and the Cr(VI) adsorption process was consistent with both Langmuir isotherm model", the word "both" should be deleted.
Author Response
Dear reviewer,
We are truly grateful to your critical comments and thoughtful suggestions. Based on these comments and suggestions, we have made careful modifications on the original manuscript. All changes made to the text are in red color. We hope the new manuscript will meet the magazine’s standard. Below you will find our point-by-point responses to your comments/questions:
If Freundlich isotherm model is deleted from the conclusion, then in line 316 "and the Cr(VI) adsorption process was consistent with both Langmuir isotherm model", the word "both" should be deleted.
√ Many thanks for your suggestion. We greatly appreciate your kind thoughts. We have corrected the mistake. Thank you very much!
Yours sincerely
Prof. Yong-Gang Zhao
College of Biological and Environmental Engineering, Zhejiang Shuren University, Hangzhou, 310015, China
E-mail address: zhyg91213@163.com